# Fracture healing is delayed in the absence of gasdermin-interleukin-1 signaling

Kai Sun[1,2,3], Chun Wang[2], Jianqiu Xiao[2], Michael D Brodt[4], Luorongxin Yuan[2], Tong Yang[1,2,3], Yael Alippe[2], Huimin Hu[3], Dingjun Hao[1,3], Yousef Abu-Amer[4,5], Matthew J Silva[4], Jie Shen[4], Gabriel Mbalaviele[2]*

[1]Xi'an Jiaotong University Health Science Center, Xi'an, China; [2]Division of Bone and Mineral Diseases, Washington University School of Medicine, St. Louis, United States; [3]Department of Spine Surgery, Honghui Hospital, Xi'an Jiaotong University, Xi'an, China; [4]Department of Orthopaedic Surgery, Washington University School of Medicine, St. Louis, United States; [5]Shriners Hospital for Children, St. Louis, United States

**Abstract** Amino-terminal fragments from proteolytically cleaved gasdermins (GSDMs) form plasma membrane pores that enable the secretion of interleukin-1β (IL-1β) and IL-18. Excessive GSDM-mediated pore formation can compromise the integrity of the plasma membrane thereby causing the lytic inflammatory cell death, pyroptosis. We found that GSDMD and GSDME were the only GSDMs that were readily expressed in bone microenvironment. Therefore, we tested the hypothesis that GSDMD and GSDME are implicated in fracture healing owing to their role in the obligatory inflammatory response following injury. We found that bone callus volume and biomechanical properties of injured bones were significantly reduced in mice lacking either GSDM compared with wild-type (WT) mice, indicating that fracture healing was compromised in mutant mice. However, compound loss of GSDMD and GSDME did not exacerbate the outcomes, suggesting shared actions of both GSDMs in fracture healing. Mechanistically, bone injury induced IL-1β and IL-18 secretion in vivo, a response that was mimicked in vitro by bone debris and ATP, which function as inflammatory danger signals. Importantly, the secretion of these cytokines was attenuated in conditions of GSDMD deficiency. Finally, deletion of IL-1 receptor reproduced the phenotype of *Gsdmd* or *Gsdme* deficient mice, implying that inflammatory responses induced by the GSDM-IL-1 axis promote bone healing after fracture.

*For correspondence:
gmbalaviele@WUSTL.EDU

## Editor's evaluation

We would like to congratulate the authors on this very exciting work that will advance our understanding on the contribution of gasdermin – interleukin-1 signaling in fracture healing. We hope future work will translate this discovery into human trials.

## Introduction

Bone fractures are one of the most frequent injuries of the musculoskeletal system. Despite advances in therapeutic interventions, delayed healing, compromised quality of the newly regenerated bone, or nonunions remain frequent outcomes of these injuries (*Clement et al., 2013*; *Muire et al., 2020*). These outcomes are complicated by the advanced age of the patients, infection, or sterile inflammation-prone comorbidities such as rheumatoid arthritis or diabetes mellitus (*Clement et al., 2013*; *Muire et al., 2020*; *Claes et al., 2012*). Although the recovery speed from fracture is greater in small animals such as rodents than in large counterparts and humans, the underlying repair mechanisms are shared

across species (*Claes et al., 2012*). Thus, mouse models, which are amenable to genetic manipulation, provide opportunities for shedding light onto the mechanisms of fracture healing.

Bone fracture triggers an immediate inflammatory response during which neutrophils and macrophages are mobilized to the injury site to remove necrotic cells and debris while releasing factors that initiate neovascularization and promote tissue repair by recruiting mesenchymal progenitor cells from various sites such as the periosteum and bone marrow (BM) (*Claes et al., 2012*; *Colnot et al., 2006*; *Glass et al., 2011*). The repair phase is followed by remodeling events where the balanced activity of osteoblasts and osteoclasts culminates in the restoration of the original bone structure and BM cavity (*Colnot et al., 2006*; *Hardy and Cooper, 2009*; *Xing et al., 2010b*). Ultimately, inflammation subsides stemming from the suppressive actions of immune cells such as regulatory T cells, anti-inflammatory macrophages, and mesenchymal stem cells (*Al-Sebaei et al., 2014*; *Nauta and Fibbe, 2007*; *Noël et al., 2007*). Although inflammation declines over time, interfering with its onset immediately after injury can be detrimental as mice lacking interleukin-6 (IL-6), tumor necrosis factor-α, or macrophages exhibit defective healing (*Yang et al., 2007*; *Wallace et al., 2011*; *Gerstenfeld et al., 2003*; *Baht et al., 2015*; *Raggatt et al., 2014*; *Xing et al., 2010a*; *Chang et al., 2008*). Thus, a fine-tuned level of inflammation is critical for adequate fracture healing.

IL-1β is another inflammatory cytokine that impacts fracture healing (*Einhorn et al., 1995*; *Morisset et al., 2007*). Unlike the aforementioned cytokines, IL-1β and IL-18 lack the signal peptide for secretion through the conventional endoplasmic reticulum and Golgi route. Expressed as pro-IL-1β and pro-IL-18, these polypeptides are proteolytically activated by enzymes such as caspase-1, a component of the intracellular macromolecular complexes called inflammasomes (*Schroder and Tschopp, 2010*; *Broz and Dixit, 2016*). Caspase-1 also cleaves GSDMD, generating GSDMD amino-terminal fragments, which form plasma membrane pores through which IL-1β and IL-18 are secreted to the extracellular milieu (*Broz and Dixit, 2016*; *Shi et al., 2015*). Although live cells can secrete these cytokines, excessive GSDMD-dependent pore formation compromises the integrity of the plasma membrane, causing a lytic form of cell death known as pyroptosis (*Shi et al., 2015*; *Evavold et al., 2018*). Pyroptotic cells release not only IL-1β and IL-18 but also other inflammatory molecules including eicosanoids, nucleotides, and alarmins (*Broz and Dixit, 2016*; *Shi et al., 2015*; *Rauch et al., 2017*). Thus, the actions of GSDMD in inflammatory settings can extend beyond the sole secretion of IL-1β and IL-18, and need to be tightly regulated to maintain homeostasis.

GSDMD is a member of the GSDM family proteins, which are encoded by *Gsdma1-3*, *Gsdmc1-4*, *Gsdmd*, and *Gsdme* also known as *Dnfa5* in the mouse genome (*Liu et al., 2021*). Mice lacking GSDMD are protected against multi-organ damage caused by gain-of-function mutations of nucleotide-binding oligomerization domain-like receptors family, pyrin domain containing 3 (NLRP3) or pyrin inflammasome (*Xiao et al., 2018*; *Kanneganti et al., 2018*). GSDMD is also involved in the pathogenesis of complex diseases including experimental autoimmune encephalitis, radiation-induced tissue injury, ischemia/reperfusion injury, sepsis, renal fibrosis, and thrombosis (*Li et al., 2019*; *Xiao et al., 2020*; *Zhang et al., 2019*; *Silva et al., 2021*; *Zhang et al., 2021c*; *Zhang et al., 2021a*). Other well-studied GSDMs include GSDMA and GSDME (*Liu et al., 2021*; *Zhang et al., 2020*; *Wang et al., 2021*; *Zhou et al., 2020*). GSDME is of particular interest to this study because recent evidence suggests that it harbors overlapping and non-overlapping actions with GSDMD, depending on cell contexts. Indeed, GSDME can mediate pyroptosis and release cytokines under both GSDMD sufficient and insufficient conditions (*Wang et al., 2021*; *Xia et al., 2021*; *Aizawa et al., 2020*; *Liu et al., 2020*; *Chen et al., 2021*). Despite advances in GSDM studies, the role that these proteins play in fracture healing has not been studied. Since drugs for the treatment of GSDM-dependent inflammatory disorders and cancers are under development, it is imperative to understand their functions in the musculoskeletal system. Here, we found that loss of GSDMD or GSDME in mice impeded fracture healing through mechanisms involving IL-1 signaling. This discovery has translational implications as drugs that inhibit GSDM functions may contribute to unsatisfactory fracture healing outcomes.

## Materials and methods
### Mice
*Gsdmd* knockout mice were kindly provided by Dr VM Dixit (Genentech, South San Francisco, CA). *Il1r1*[-/-] and *Gsdme*[-/-] mice were purchased from The Jackson Laboratory (Sacramento, CA). All mice

were on the C57BL6J background, and genotyping was performed by PCR. All procedures were approved by the Institutional Animal Care and Use Committee (IACUC) of Washington University School of Medicine in St Louis. All experiments were performed in accordance with the relevant guidelines and regulations described in the IACUC-approved protocol 19-0971.

### Tibia fracture model

Open mid-shaft tibia fractures were created unilaterally in 12-week-old mice. Briefly, a 6-mm-long incision was made in the skin on the anterior side alongside the tibia. A sterile 26 G needle was inserted into the tibia marrow cavity from the proximal end, temporarily withdrawn to allow transection of the tibia with a scalpel at mid-shaft, and then reinserted and secured. The incision was closed with 5–0 nylon sutures. Mice were sacrificed at different time-points as indicated below.

### Histological analyses of fracture calluses

Fractured tibias were collected on days 7, 10, 14, 21, and 28 after fracture for histological analyses. Excess muscle and soft tissue were excised. Tibias were fixed in 10% neutral buffered formalin for 24 hr and decalcified for 10–14 days in 14% ethylenediaminetetraacetic acid solution (pH 7.2). Tissue was processed and embedded in paraffin, and sectioned longitudinally at a thickness of 5 μm. Alcian blue/hematoxylin/orange-g (ABH/OG) and tartrate-resistant acid phosphatase (TRAP) staining were performed to analyze the callus composition and osteoclast formation in the fracture region. Images were acquired using ZEISS microscopy (Carl Zeiss Industrial Metrology, Maple Grove, MN). Cartilage area, bone area, mesenchyme area, and osteoclast parameters were quantified on ABH/OG, TRAP-stained sections using NIH ImageJ software 1.52a (Wayne Rasband) and Bioquant (*Ying et al., 2020*).

### Micro-computed tomography analysis

After careful dissection and removal of the intramedullary pins in fractured tibias, fracture calluses were examined using micro-computed tomography (micro-CT) system (μCT 40 scanner, Scanco Medical AG, Zurich) scanner at 10 μm, 55 kVp, 145 μA, 300 ms integration time. Six hundred slices (6.3 mm) centered on the callus midpoint were used for micro-CT analyses. A contour was drawn around the margin of the entire callus and a lower threshold of 180 per mille was then applied to segment mineralized tissue (all bone inside the callus). A higher threshold of 340 per mille was applied to segment the original cortical bone inside the callus volume. Quantification for the volumes of the bone calluses was performed using the Scanco analysis software. 3D images were generated using a constant threshold of 180 per mille for the diaphyseal callus region of the fractured tibia.

### Biomechanical torsion testing

Tibias were collected 28 days after fracture and moistened with PBS and stored at –20°C until they were thawed for biomechanical testing. Briefly, the ends of the samples were potted with methacrylate (MMA) bone cement (Lang Dental Manufacturing, Wheeling, IL) in 1.2-cm-long cylinders (6 mm diameter). The fracture site was kept in the center of the two potted ends with roughly 4.2 mm of the bone exposed. After MMA solidification, potted bones were set up on a custom torsion machine. One end of the potted specimen was held in place while the opposing end was rotated at 1 degree per second until fracture. Torque values were plotted against the rotational deformation, and the maximum torque, torsional rigidity, and work to fracture were calculated.

### Cell cultures

Murine primary BM-derived macrophages (BMDMs) were obtained by culturing mouse BM cells from femurs and tibias in culture media containing a 1:10 dilution of supernatant from the fibroblastic cell line CMG 14-12 as a source of macrophage colony-stimulating factor, a mitogenic factor for BMDMs, for 4–5 days in a 15 cm dish as previously described (*Xiao et al., 2020*; *Takeshita et al., 2000*). After expansion, BMDMs were plated at a density of $1 \times 10^6$ cells/well in six-well plate for experiments.

Murine primary neutrophils were isolated by collecting BM cells and subsequently over a discontinuous Percoll (Sigma) gradient. Briefly, all BM cells from femurs and tibias were washed by DPBS and then resuspend in 2 ml DPBS. Cell suspension was gently layered on top of gradient (72% Percoll, 64% Percoll, 52% Percoll) and centrifuged at 1545× *g* for 30 min at room temperature. After carefully discarding the top two cell layers, the third layer containing neutrophils was transferred to a clean

15 ml tube. Cells were washed and counted, then plated at a density of $3 \times 10^6$ cells/well in six-well plate. Neutrophil purity was determined by flow cytometry shown in *Figure 5—figure supplement 1*.

For inflammasome studies, cells were primed with 100 ng/ml LPS (Sigma Aldrich, L4391) for 3 hr, then with 15 µM nigericin (Sigma Aldrich) for 1 hr, 5 mM ATP for 1 hr, or 50 mg/ml bone particles for 2 hr.

## Western blot

Cell extracts were prepared by lysing cells with RIPA buffer (50 mM Tris, 150 mM NaCl, 1 mM EDTA, 0.5% NaDOAc, 0.1% SDS, and 1.0% NP-40) plus complete protease inhibitor cocktail (Roche, CA). For tissue extracts, BM and BM-free bones were lysed with RIPA buffer containing protease inhibitors. Protein concentrations were determined by the Bio-Rad Laboratories method, and equal amounts of proteins were subjected to SDS-PAGE gels (12%) as previously described (*Wang et al., 2018*). Proteins were transferred onto nitrocellulose membranes and incubated with antibodies against GSDMD (1:1000, ab219800, Abcam), GSDME (1:1000, ab215191, Abcam), β-actin (1:5000, sc-47778, Santa Cruz Biotechnology, Dallas, TX) overnight at 4°C followed by incubation for 1 hr with secondary goat anti–mouse IRDye 800 (Thermo Fisher Scientific, Waltham, MA) or goat anti-rabbit Alexa Fluor 680 (Thermo Fisher Scientific, Waltham, MA), respectively. The results were visualized using the Odyssey infrared imaging system (LI-COR Biosciences, Lincoln, NE).

## LDH assay

Cell death was assessed by the release of LDH in conditioned medium using LDH Cytotoxicity Detection Kit (TaKaRa, San Jose, CA).

## IL-1β and IL-18 ELISA

IL-1β, IL18 levels in conditioned media were measured by ELISA (eBioscience, Albany, NY).

## ATP assay

ATP levels in conditioned media were measured by RealTime-Glo Extracellular ATP Assay kit (Promega, Madison, WI).

## Flow cytometry

BM cells were flushed from tibias with PBS. Single cell suspensions were labeled with antibodies for 30 min at 4°C. Flow cytometry analysis was performed on FACS Canto II. Cell cytometric data was analyzed using FlowJo10.7.1. Full gating strategy was shown in *Figure 5—figure supplements 1 and 2*.

## RNA isolation and RT-qPCR

RNA was extracted from bone or BM cells using RNeasy Plus Mini Kit (Qiagen). Four millimeters of fracture calluses free of BM were homogenized for mRNA extraction. cDNA were prepared using High-Capacity cDNA Reverse Transcription Kits (Applied Biosystems, Waltham, MA). Gene expression was analyzed by qPCR using SYBR Green (Applied Biosystems, Waltham, MA) according to the manufacturer's instruction.

## Statistical analysis

Statistical analysis was performed using the Student's t test, one-way ANOVA with Tukey's multiple comparisons test as well as two-way ANOVA with Tukey's multiple comparisons test in GraphPad Prism 8.0 software.

# Results

## GSDMD and GSDME were expressed in bone microenvironment

The crucial role that gasdermins (GSDMs) play in inflammation, a response that can be induced by injury, prompted us to analyze their expression in unfractured and fractured mouse tibias. *Gsdmd* and *Gsdme* were the only GSDM family members that were readily detected in BM and BM-free tibias from wild-type (WT) mice (*Figure 1A–D* and *Figure 1—figure supplement 1A, B*; *Figure 1—figure*

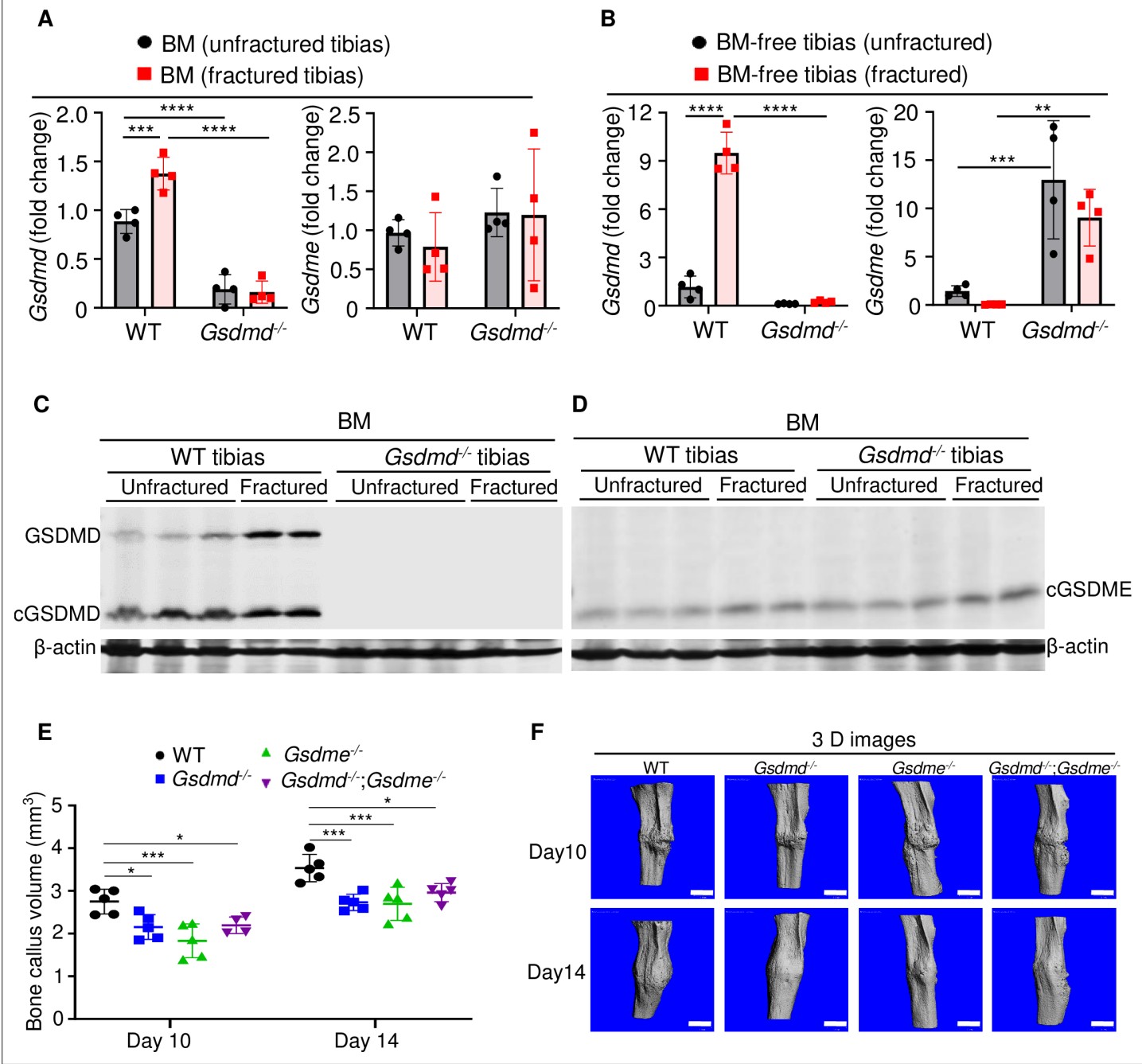

**Figure 1.** GSDMD and GSDME were expressed in bone microenvironment and involved in bone callus formation. (**A, C–D**) BM and (**B**) BM-free tibias from 12-week-old male WT and *Gsdmd*-/- mice (n = 4–5 mice). Samples were isolated from unfractured or fractured tibias (3 days after injury). (**A–B**) qPCR and (**C–D**) Western blot analyses. qPCR data were normalized to unfractured WT. (**E**) Bone callus volume was quantified using Scanco software (n = 5). (**F**) Representative 3D reconstructions of bones using μCT. Data were mean ± SD and are representative of at least three independent experiments. Data from male and female mice were pooled because there was no sex difference. **p < 0.01; ***p < 0.001; ****p < 0.0001, two-way ANOVA with Tukey's multiple comparisons test. Scale bar, 1 mm. BM, bone marrow; μCT, micro-computed tomography; WT, wild-type.

The online version of this article includes the following source data and figure supplement(s) for figure 1:

**Source data 1.** qPCR analysis of *Gsdmd* and *Gsdme* expression in bone marrow (BM) and BM-free tibias of wild-type (WT) and *Gsdmd*-/- mice.

**Source data 2.** Western blots for *Figure 1*.

**Source data 3.** Bone callus volume of wild-type (WT), *Gsdmd*-/-, *Gsdme*-/-, and *Gsdmd*-/-;*Gsdme*-/- mice.

**Figure supplement 1.** Several gasdermin (GSDM) family members were barely expressed in bones.

*Figure 1 continued on next page*

*Figure 1 continued*

**Figure supplement 1—source data 1.** qPCR analysis of *Gsdm* expression in bone marrow (BM) and BM-free tibias of wild-type (WT) and *Gsdmd*$^{-/-}$ mice.

**Figure supplement 2.** Fracture-induced GSDMD and GSDME expression.

**Figure supplement 2—source data 1.** qPCR analysis of gene expression in bone marrow (BM) and BM-free tibias of wild-type (WT) and *Gsdmd*$^{-/-}$ mice.

**Figure supplement 2—source data 2.** Western blots for *Figure 1—figure supplement 2A and B*.

supplement 2A, B). Expression levels of *Gsdmd* in BM and BM-free tibias (*Figure 1A–C* and *Figure 1—figure supplement 2A*) were consistently higher in fractured compared with unfractured bones. The injury did not affect *Gsdme* mRNA levels (*Figure 1A*) but it increased GSDME protein levels in BM-free tibias (*Figure 1—figure supplement 2B*). Both GSDMs appeared constitutively cleaved in BM (*Figure 1C–D*) but not BM-free tibias (*Figure 1—figure supplement 2A, B*). Since *Gsdmd* was predominantly expressed in bones, we determined the impact of its loss on the expression of its family members. *Gsdmd* deficiency increased baseline *Gsdme* mRNA levels in BM-free tibias but not BM, a response that was unaffected by the injury and did not impact GSDME protein levels (*Figure 1A–D* and *Figure 1—figure supplement 2A, B*). The expression of the other family members was unaltered by *Gsdmd* deficiency or the injury, with exception of *Gsdmc*, which was reduced in fractured BM-free tibias (*Figure 1—figure supplement 1A, B*). Thus, GSDMD and GSDME are present in the bone microenvironment in homeostatic and injury states.

## Lack of GSDMD or GSDME delayed fracture healing

When stabilized with an intramedullary inserted pin, fractured murine long bones heal through mechanisms that involve the formation of callus structures (*Marsell and Einhorn, 2011*). To determine the role of GSDMD and GSDME in fracture healing, we assessed callus formation in WT, *Gsdmd*$^{-/-}$, *Gsdme*$^{-/-}$, and *Gsdmd*$^{-/-}$;*Gsdme*$^{-/-}$ mice. The volume of bone callus was higher on day 14 compared with day 10 post-injury in WT mice (*Figure 1E*). It increased indistinguishably in *Gsdmd*$^{-/-}$ and *Gsdme*$^{-/-}$ mice on day 14 compared to day 10, but was significantly lower in mutants compared with WT mice (*Figure 1E–F*). Notably, callus volume was comparable between single and compound mutants (*Figure 1E–F*), suggesting that both GSDMs share the same signaling pathway in fracture healing. Collectively, these findings indicate that GSDMD and GSDME play an important role in bone repair following fracture injury.

To gain insights onto the mechanisms of fracture healing, we focused on GSDMD as its expression and proteolytic maturation were consistently induced by fracture. Time-course studies revealed that while the callus bone volume increased linearly until day 14 post-fracture and plateaued by day 21 in WT mice (*Figure 2A–B*), it continued to expand in *Gsdmd*$^{-/-}$ mice until day 21 (*Figure 2A–B*). The callus volume declined in both mouse strains by day 28 but it was larger in mutants compared with WT controls (*Figure 2A*). Histological analysis indicated that the areas of the newly formed mesenchyme, cartilage, and bone were smaller in *Gsdmd*$^{-/-}$ compared with WT mice on day 7 (*Figure 2C–F*). This outcome was also observed on day 10, except for the mesenchyme tissue area, which was larger in *Gsdmd*$^{-/-}$ compared with WT. While cartilage remnants were negligible in WT callus on day 14, they remained abundant in *Gsdmd*$^{-/-}$ counterparts (*Figure 2F*). Additional histological assessments revealed that the number osteoclasts, which are involved in the remodeling of the newly formed bone, declined after day 14 not only in WT bones as expected, but also in mutants (*Figure 2G–I*). At any time-point, there were more osteoclasts in injured *Gsdmd*$^{-/-}$ bones compared to WT counterparts. Taken together, these results suggest that the healing process is perturbed in mutant mice.

To determine the impact of GSDMD deficiency on the functional result of bone regeneration, unfractured and 28 days' post-fracture bones were subjected to biomechanical testing. Injured WT tibias exhibited decreased strength and stiffness compared with unfractured counterparts (*Figure 3A–C*), indicating that the healing response has not fully recovered bone function at this time-point. Biomechanical properties of unfractured *Gsdmd*$^{-/-}$ tibias were slightly higher though not statistically significant in *Gsdmd*$^{-/-}$ compared with WT unfractured bones (*Figure 3A–C*), findings that were consistent with the higher bone mass phenotype of *Gsdmd*$^{-/-}$ mice relative to their littermates (*Xiao et al., 2020*). Notably, fractured bones from *Gsdmd*$^{-/-}$ mice exhibited lower biomechanical parameters compared

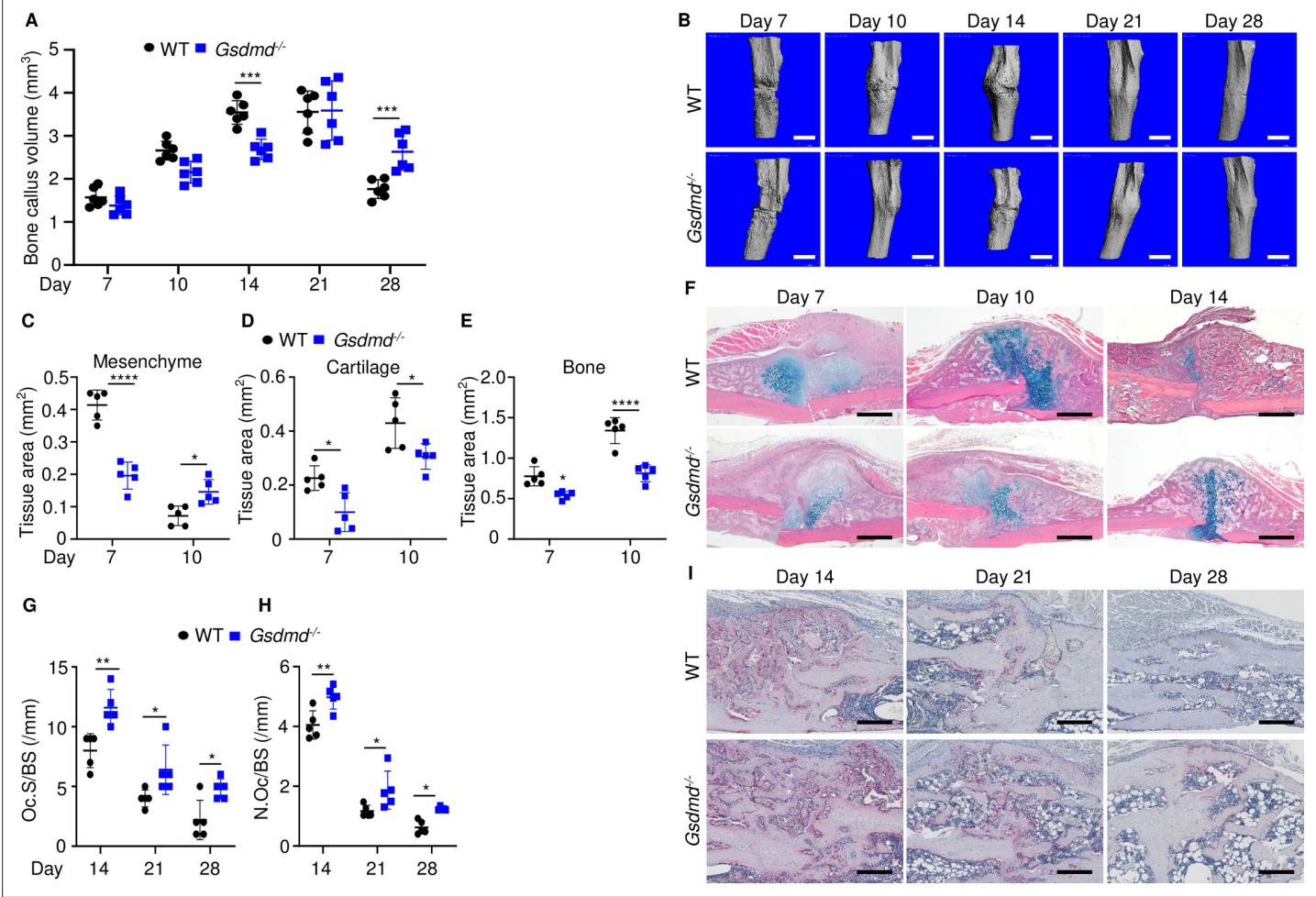

**Figure 2.** Loss of GSDMD delayed fracture healing. Tibias of 12-week-old male and female WT or *Gsdmd*⁻/⁻ mice were subjected to fracture and analyzed at the indicated times. (**A**) Bone callus volume was quantified using Scanco software (n = 6). (**B**) Representative 3D reconstructions of bones using µCT. (**C–E**) Quantification of tissue area by ImageJ software (n = 5). (**F**) Representative ABH staining. Quantification of Oc.S/BS (**G**) and N.Oc/BS (**H**) using Bioquant software (n = 5). (**I**) Representative images of TRAP staining. Data were mean ± SD. Data from male and female mice were pooled because there was no sex difference. *p < 0.05; ***p < 0.001; ****p < 0.0001, two-way ANOVA with Tukey's multiple comparisons test. Scale bar, 1 mm (**B**), 500 µm (**F**) ,or 200 µm (**I**). µCT, micro-computed tomography; WT, wild-type.

The online version of this article includes the following source data for figure 2:

**Source data 1.** Micro-computed tomography (µCT) and histological, and histomorphometric analyses of wild-type (WT) and *Gsdmd*⁻/⁻ mice.

with WT controls. Thus, the functional competence of the repaired bone structure is compromised in GSDMD-deficient mice.

## Expression and secretion of IL-1β and IL-18 were attenuated in the absence of GSDMD

Inflammation characterized by elevated levels of cytokines including those of the IL-1 family underlines the early phase of wound healing (*Claes et al., 2012*). Since IL-1β and IL-18 are secreted through GSDMD-assembled plasma membrane pores (*Broz and Dixit, 2016*; *Shi et al., 2015*; *Evavold et al., 2018*), we analyzed the levels of these inflammatory cytokines in the BM supernatants from unfractured and fractured bones (1 day after injury). Baseline secretion levels of IL-1β or IL-18 were comparable between WT and *Gsdmd* mutants (*Figure 4A–B*). Fracture increased IL-1β and IL-18 levels in BM supernatants in both groups, but they were significantly attenuated in mutant samples compared with WT controls (*Figure 4A–B*). Thus, fracture-induced IL-1β and IL-18 levels in BM are attenuated upon loss of GSDMD.

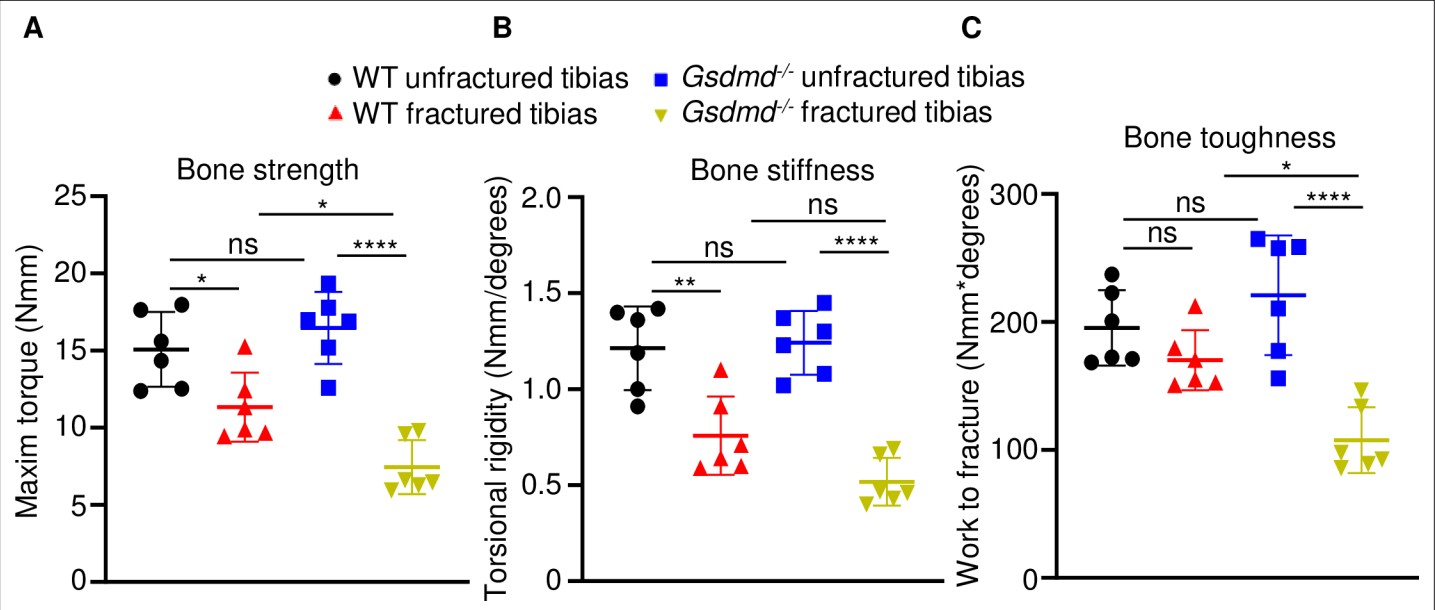

**Figure 3.** Loss of GSDMD compromised bone biomechanical properties after fracture. Unfractured or fractured tibias (28 days after injury) from 12-week-old male WT or *Gsdmd*⁻/⁻ mice were subjected to a torsion test (n = 6). (**A**) Bone strength. (**B**) Bone stiffness. (**C**) Bone toughness. Data were mean ± SD. *p < 0.05; **p < 0.01; ****p < 0.0001; one-way ANOVA with Tukey's multiple comparisons test; ns, non significant; WT, wild-type.

The online version of this article includes the following source data for figure 3:

**Source data 1.** Biomechanical analysis of wild-type (WT) and *Gsdmd*⁻/⁻ mice.

To understand transcriptional regulation of IL-1β and IL-18 in this fracture model, we determined mRNA levels of these cytokines in the BM and BM-free bone compartments. Baseline levels of *Il1b* and *Il18* mRNA were undistinguishable between WT and *Gsdmd*⁻/⁻ samples in both compartments (*Figure 4C–D*). Following fracture, the expression of *Il1b* and *Il18* mRNA was induced in WT but not *Gsdmd*⁻/⁻ mice (*Figure 4C–D*), suggesting a feedback mechanism whereby these cytokines secreted though GSDMD pores amplified their own expression. Since NLRP3 and absent in melanoma 2 (AIM2) inflammasomes, which sense plasma stimuli such as membrane perturbations and DNA, respectively, are implicated in the maturation of IL-1β, IL-18, and GSDMD (*Xiao et al., 2018*; *Xiao et al., 2020*; *Zhu et al., 2021*), we also analyzed the expression of these sensors. Levels of *Nlrp3* and *Aim2* to some extent (*Figure 4C–D*) as well as those of *Asc* and *caspase-1* (*Figure 4—figure supplement 1A, B*) were comparable between WT and *Gsdmd*⁻/⁻ samples in homeostatic conditions. Fracture increased the expression of *Nlrp3* and *Aim2* in WT and mutants only in BM-free bone samples (*Figure 4C–D*) whereas it induced caspase-1 expression in WT cells both compartments. Never was the expression of *Nlrc4* and *caspase-11* mRNA modulated by the fracture injury nor loss of GSDMD (*Figure 4—figure supplement 1A, B*). Thus, the expression of *Il1b* or *Il18* and certain inflammasome components (e.g., *Nlrp3*, *Aim2*, *Asc*, and *caspase-1*) is transcriptionally regulated in the fracture injury model.

## Lack of GSDMD attenuated the secretion of IL-1β and IL-18 induced by danger signals

The high levels IL-1β and IL-18 in BM of fractured bones provided a strong rationale for assessing the presence of neutrophils, monocytes, and macrophages, which harbor high levels of inflammasomes and rapidly accumulate during the first hours after injury (*Liu et al., 2021*; *Xiao et al., 2018*; *Xiao et al., 2020*). Flow cytometry analysis revealed that the abundance of these cells in BM of uninjured bones was unaffected by loss of GSDMD (*Figure 5A–C* and *Figure 5—figure supplement 2*). Fracture increased the percentage of neutrophils and monocytes but not macrophages (*Figure 5A–C*). GSDMD deficiency was associated with a slight decrease and increase in the percentage of neutrophils and monocytes, respectively (*Figure 5A–C*). Thus, neutrophil and monocyte but not macrophage populations are expanded in fractured bones. GSDMD deficiency appears to slightly attenuate and increase the percentage of neutrophils and monocytes, respectively.

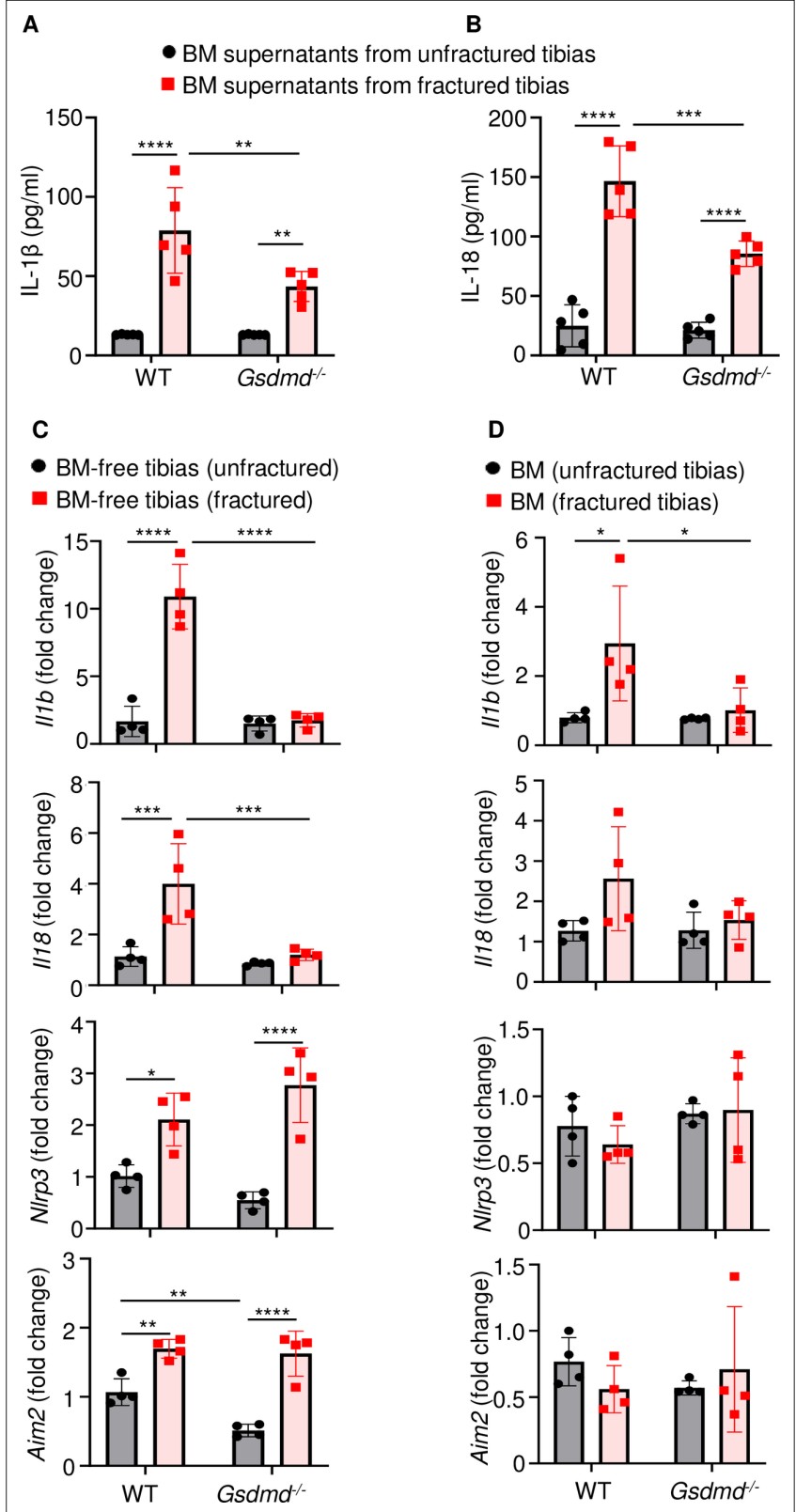

**Figure 4.** Loss of GSDMD attenuated the expression and secretion of interleukin-1β (IL-1β) and IL-18 induced by fracture. (**A–B**) BM supernatants and (**C**) BM-free bones were from 12-week-old male WT or *Gsdmd*-/- mice (n = 4–5). Samples were isolated from unfractured or fractured tibias (1 day after injury). (**A–B**) ELISA and (**C–D**) qPCR analyses. qPCR data were normalized to unfractured WT. Data were mean ± SD. *p < 0.05; **p < 0.01;

*Figure 4 continued on next page*

*Figure 4 continued*

***p < 0.001; ****p < 0.0001, two-way ANOVA with Tukey's multiple comparisons test. BM, bone marrow; WT, wild-type.

The online version of this article includes the following source data and figure supplement(s) for figure 4:

**Source data 1.** ELISA analysis of interleukin-1β (IL-1β) and IL-18 levels in bone marrow (BM) supernatants of wild-type (WT) and *Gsdmd*[-/-] mice.

**Source data 2.** qPCR analysis of gene expression in bone marrow (BM) and BM-free tibias of wild-type (WT) and *Gsdmd*[-/-] mice.

**Figure supplement 1.** Fracture induced the expression of certain inflammasome components.

**Figure supplement 1—source data 1.** Percentage of lactate dehydrogenase (LDH) release in wild-type (WT) and *Gsdmd*[-/-] cell culture supernatants.

Inflammasome assembly signals include those generated by ATP, which is released by dead cells (*Yang et al., 2021*; *Zhang and Wei, 2021b*). Therefore, we measured the levels of this danger signal in BM. ATP levels were comparable between WT and *Gsdmd*[-/-] samples at baseline but were induced by fourfold after fracture in both groups (*Figure 5D*). Next, we studied cytokine release by WT and *Gsdmd*[-/-] cells not only in response to ATP but also bone particles, which are undoubtedly released following bone fracture. Bone particles were as potent as the NLRP3 inflammasome activators, nigericin and ATP, in inducing GSDMD cleavage by LPS-primed macrophages (*Figure 5E*). Accordingly, these danger signals induced IL-1β release (*Figure 5F*) and pyroptosis as assessed by the release of lactate dehydrogenase (LDH; *Figure 5—figure supplement 3A*), responses that were attenuated in GSDMD-deficient macrophages. Both nigericin and ATP robustly stimulated GSDMD cleavage and IL-1β release by LPS-primed neutrophils through mechanisms that partially involved GSDMD, but they did not promote neutrophil pyroptosis (*Figure 5—figure supplement 3B*). Bone particles had no effect on GSDMD maturation and IL-1β and LDH release by neutrophils (*Figure 5G–H* and *Figure 5—figure supplement 3B*). Thus, fracture injury creates a microenvironment that induces cytokine secretion through mechanisms involving GSDMD.

## Loss of IL-1 signaling delayed fracture healing

The inability of *Gsdmd*[-/-] mice to mount efficient healing responses correlated with low levels of IL-1β and IL-18 in BM, suggesting that inadequate secretion of these cytokines may account for the delayed fracture repair. While the actions of IL-18 in bone are not well defined, overwhelming evidence positions IL-1β as a key regulator of skeletal pathophysiology (*Mbalaviele et al., 2017*; *Novack and Mbalaviele, 2016*). Therefore, we used IL-1 receptor knockout (*Il1r1*[-/-]) mice to test the hypothesis that IL-1 signaling was required for bone healing following fracture. Bone callus volume was larger on day 14 compared with day 10 in WT and *Il1r1*[-/-] tibias, but it was smaller at both time-points in mutants compared to WT controls (*Figure 6A–B*). Histological analysis confirmed that the mesenchyme, cartilage, and bone areas were all smaller in *Il1r1*[-/-] compared with WT mice (*Figure 6C–F*). Like in *Gsdmd*[-/-] tissues, cartilage remnants were prominent within the callus of *Il1r1*[-/-] specimens at day 14 (*Figure 6F*), and the number and surface of osteoclasts were significantly higher at all times in mutant compared to WT mice at day 14, while cartilage and bone areas remained smaller in mutants at day 10 (*Figure 6G–I*). Although Gsdmd expression was not analyzed in *Il1r1*[-/-] mice, these mutants exhibited delayed fracture healing like *Gsdmd*[-/-] mice, suggesting that functional GSDMD-IL-1 axis is important for adequate bone healing after fracture.

## Discussion

GSDMs are implicated in a variety of inflammatory diseases but their role in bone regeneration after fracture is largely unknown. We found that fracture healing was comparably delayed in mice lacking GSDMD or GSDME; yet concomitant loss of GSDMD and GSDME did not worsen the phenotype. This outcome was unexpected because these GSDMs are activated via distinct mechanisms as GSDMD is cleaved by caspase-1, caspase-11 (mouse ortholog of human caspase-4 and -5), neutrophil elastase and cathepsin G, or caspase-8 whereas GSDME is processed by caspase-3 and granzyme B (*Zhang et al., 2020*; *Liu et al., 2020*). The phenotype of double knockout mice suggested complex and

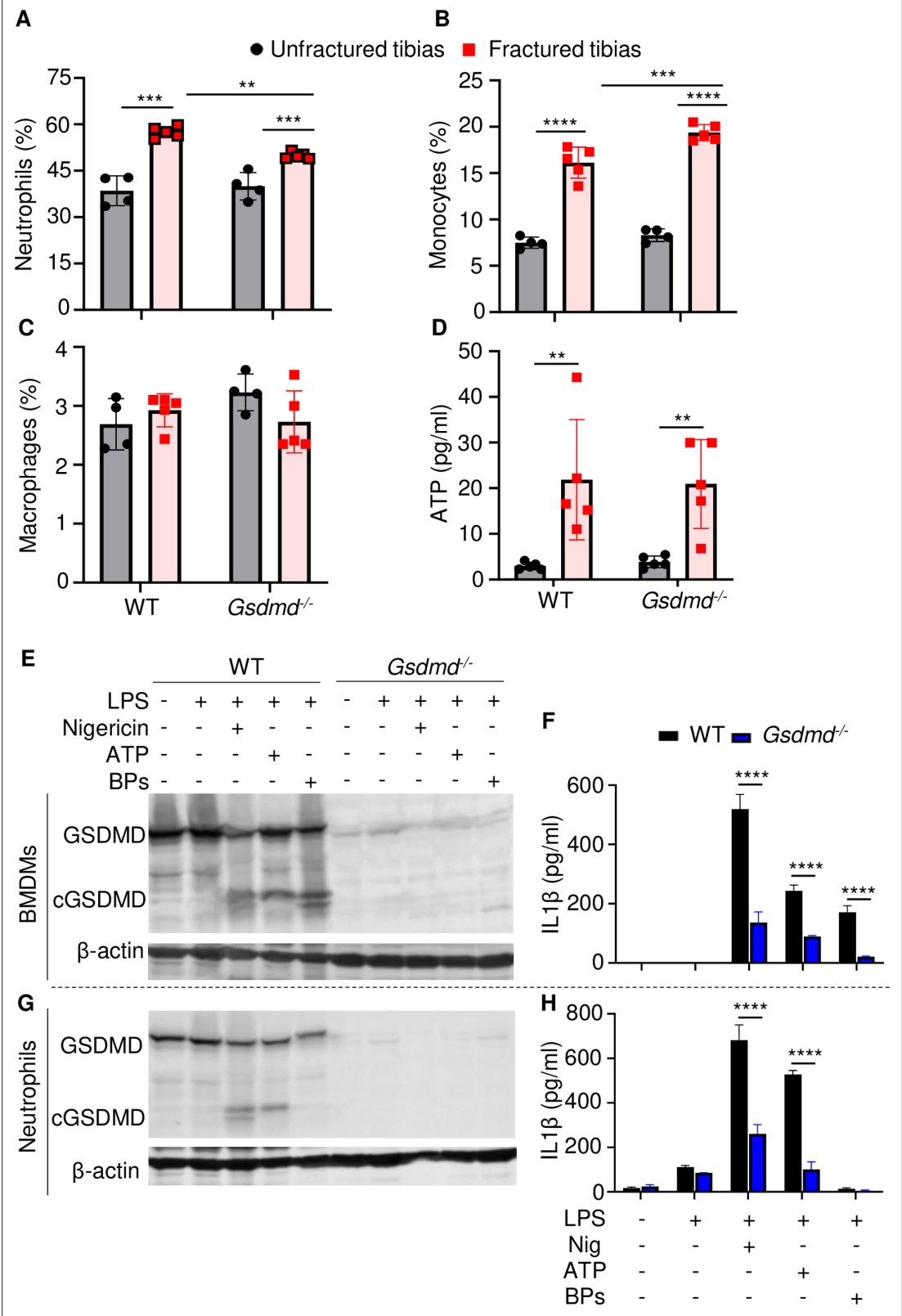

**Figure 5.** Loss of GSDMD attenuated the secretion of interleukin-1β (IL-1β) and IL-18 induced by danger signals. Cells were isolated from the tibias from 12-week-old female WT or *Gsdmd⁻/⁻* mice. (**A–C**) Cell counts (n = 4–5). BM was harvested from unfractured or fractured tibias (2 days after fracture). (**D**) ATP levels. BM supernatants were harvested from unfractured or fractured tibias (24 hr after fracture). (**E–G**) Immunoblotting analysis of GSDMD cleavage or (**F–H**) IL-1β ELISA run in triplicates. Bone marrow-derived macrophages (BMDMs) were expanded in vitro whereas neutrophils were

*Figure 5 continued on next page*

Figure 5 continued

immediately after purification. Cells were primed with 100 ng/ml LPS for 3 hr, then with 15 μM nigericin for 1 hr, 5 mM ATP for 1 hr, or 50 mg/ml bone particles for 2 hr. Data are mean ± SD and were representative of at least three independent experiments. *p < 0.05; **p < 0.01; ***p < 0.001; ****p < 0.0001, two-way ANOVA with Tukey's multiple comparisons test. BPs, bone particles; cGSDMD, cleaved GSDMD; WT, wild-type.

The online version of this article includes the following source data and figure supplement(s) for figure 5:

**Source data 1.** ELISA analysis of interleukin-1β (IL-1β) and IL-18 levels in wild-type (WT) and *Gsdmd*$^{-/-}$ cell culture supernatants.

**Source data 2.** Micro-computed tomography (μCT) and histological, and histomorphometric analyses of wild-type (WT) and Il-1r$^{-/-}$ mice.

**Figure supplement 1.** Gating strategy and purity of isolated neutrophil fractions.

**Figure supplement 2.** Gating strategy.

**Figure supplement 3.** Effects of GSDMD loss on LDH release.

possibly convergent actions of these GSDMs in fracture healing. Indeed, the phenotype of either single mutant strain implied non-redundant functions of both GSDMs whereas the outcomes of compound mutants suggested that they shared downstream effector molecules. The latter view was supported by the ability of either GSDM to mediate pyroptosis and IL-1β and IL-18 release in cell a context-dependent manner (*Wang et al., 2021*; *Xia et al., 2021*; *Aizawa et al., 2020*; *Liu et al., 2020*; *Chen et al., 2021*). Furthermore, functional complementation of GSDMD by GSDME has also been reported (*Wang et al., 2021*). Thus, although additional studies are needed for further insights onto the mechanism of differential actions of GSDMD and GSDME in bone recovery after injury, our findings reinforce the crucial role that inflammation plays during the bone fracture healing process.

In addition to IL-1β and IL-18, inflammatory mediators such as ATP, alarmins (e.g., IL-1α, S100A8/9), high mobility group box 1, and eicosanoids (e.g., PGE$_2$) are expected to be uncontrollably released during pyroptosis (*Rauch et al., 2017*; *Nyström et al., 2013*). Since these inflammatory and danger signals work in concert to inflict maximal tissue damage, we anticipated a milder delay in fracture healing in mice lacking IL-1 receptor compared with GSDM-deficient mice. Contrary to our expectation, callus volumes and the recovery time were comparable among all mutant mouse strains. These observations suggested that IL-1 signaling downstream of GSDMs played a non-redundant role in fracture healing. This view was consistent with the reported essential role of IL-1α and IL-1β in bone repair and aligned with the high expression of these cytokines by immune cells, which are known to massively infiltrate the fracture site (*Claes et al., 2012*). Thus, although IL-1 signaling has been extensively studied in various injury contexts, the novelty of this work is its demonstration of the role of GSDM-IL-1 axis in fracture repair.

IL-1 signaling induces the expression of cytokines, chemokines, and growth factors that govern bone remodeling, a process that is initiated by the osteoclasts (*Kitazawa et al., 1994*). Consistent with the critical actions of GSDM-IL-1 cascades in bone repair and the osteoclastogenic actions of IL-1β, loss of GSDMD in mice resulted in increased bone mass at baseline as the result of decreased osteoclast differentiation (*Xiao et al., 2020*). Here, we found that lack of GSDMD was associated with increased number of osteoclasts and their precursors, the monocytes. We surmised that this result was simply a reflection of delayed osteoclastogenesis in mutant mice as the consequence of delayed responses such as neovascularization and development of BM cavity. This view is based by the fact that BM is the site of hematopoiesis in adult animals and vascularization is important for the traffic of osteoclast progenitors. As a result, disruption in either hematopoiesis or vascularization should undoubtedly impact osteoclastogenesis (*Buettmann et al., 2019*). Although our mechanistic studies revolved around myeloid cells from which the osteoclasts arise, it is worth noting that the inflammasome-GSDM pathways are functional in the osteoblast lineage (*Zhang and Wei, 2021b*; *Lei et al., 2021*; *Jiang et al., 2021*). Therefore, osteoblast lineage autonomous actions of these pathways in bone healing cannot be ruled out.

Fracture injury increased the levels of NLRP3 and its activator ATP in BM, suggesting that the inflammasome assembled by this sensor may be responsible for the activation of GSDMs, particularly, GSDMD. However, more studies are needed to firm up this conclusion since Aim2 was also expressed in BM. Other limitations of this study included (i) the lack of data comparing biomechanical properties of fractured bones from all the mutant mouse strains used; (ii) the unknown function of IL-18 pathway in fracture healing; (iii) the contribution of systemic factors such as glucocorticoids, which are regulated by inflammasome-IL-1β pathways and affect bone healing and strength (*Stangl et al.,*

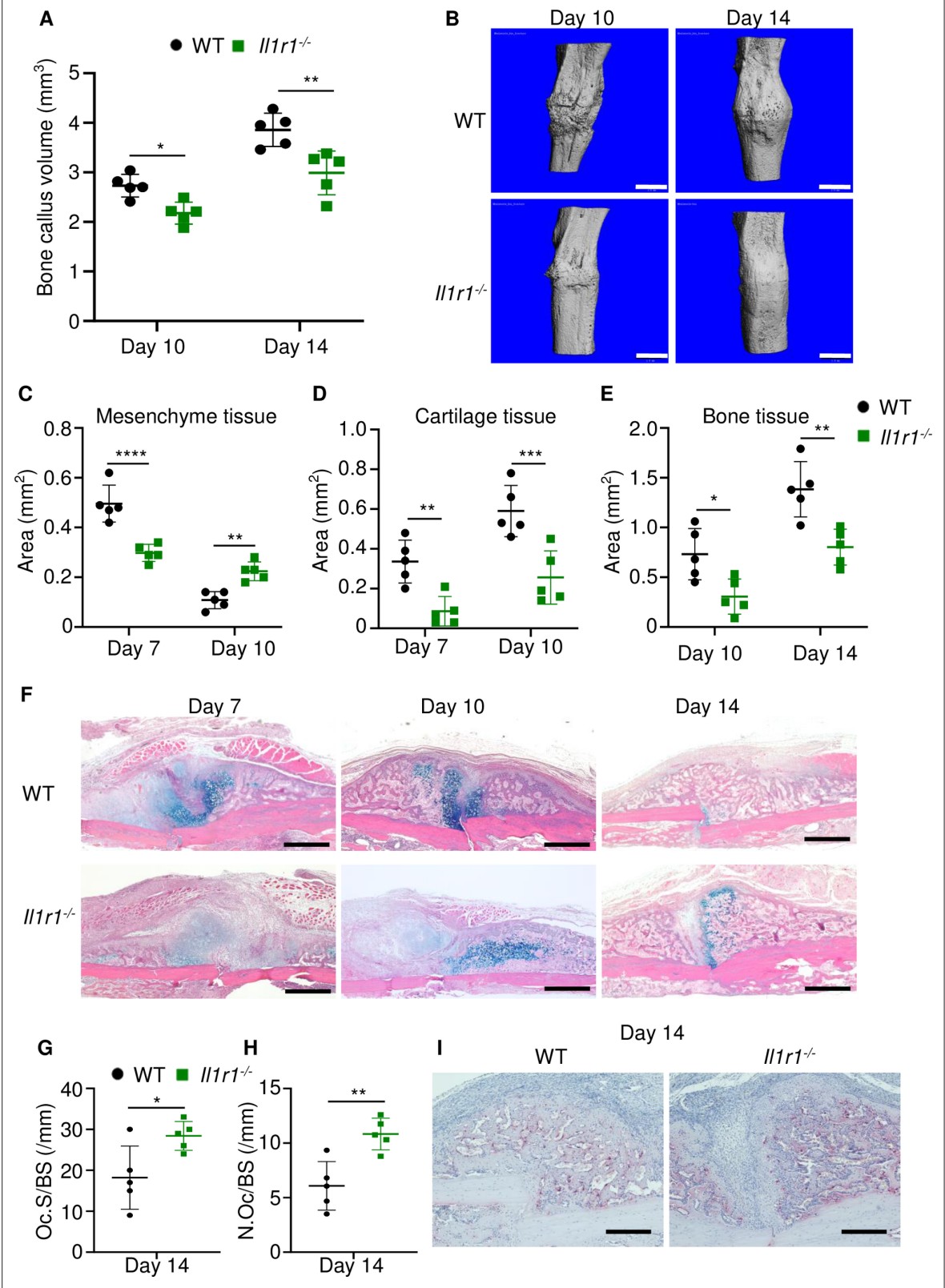

**Figure 6.** Loss of interleukin-1 (IL-1) receptor delayed fracture healing. Tibias of 12-week-old male WT or *Il1r1⁻/⁻* mice were subjected to fracture and analyzed at the indicated times. (**A**) Bone callus volume was quantified using Scanco software (n = 5). (**B**) Representative 3D reconstructions of bones using micro-computed tomography (μCT). (**C–E**) Quantification of tissue area by ImageJ software (n = 5). (**F**) Representative ABH staining. (**G**) Quantification of Oc.S/BS and (**H**) Oc.S/BS using Bioquant software (n = 5). (**I**) Representative images of TRAP staining. Data were mean ± SD.

*Figure 6 continued on next page*

*Figure 6 continued*

*p < 0.05; ***p < 0.001; ****p < 0.0001, two-way ANOVA with (**A, C–E**) Tukey's multiple comparisons test or (**G–H**) unpaired t-test. Scale bar, 1 mm (B), 500 µm, (F) or 200 µm (I). *Il1r1*, IL-1 receptor 1; WT, wild-type.

The online version of this article includes the following source data for figure 6:

**Source data 1.** Callus parameters of WT and *Il1r⁻/⁻* mice.

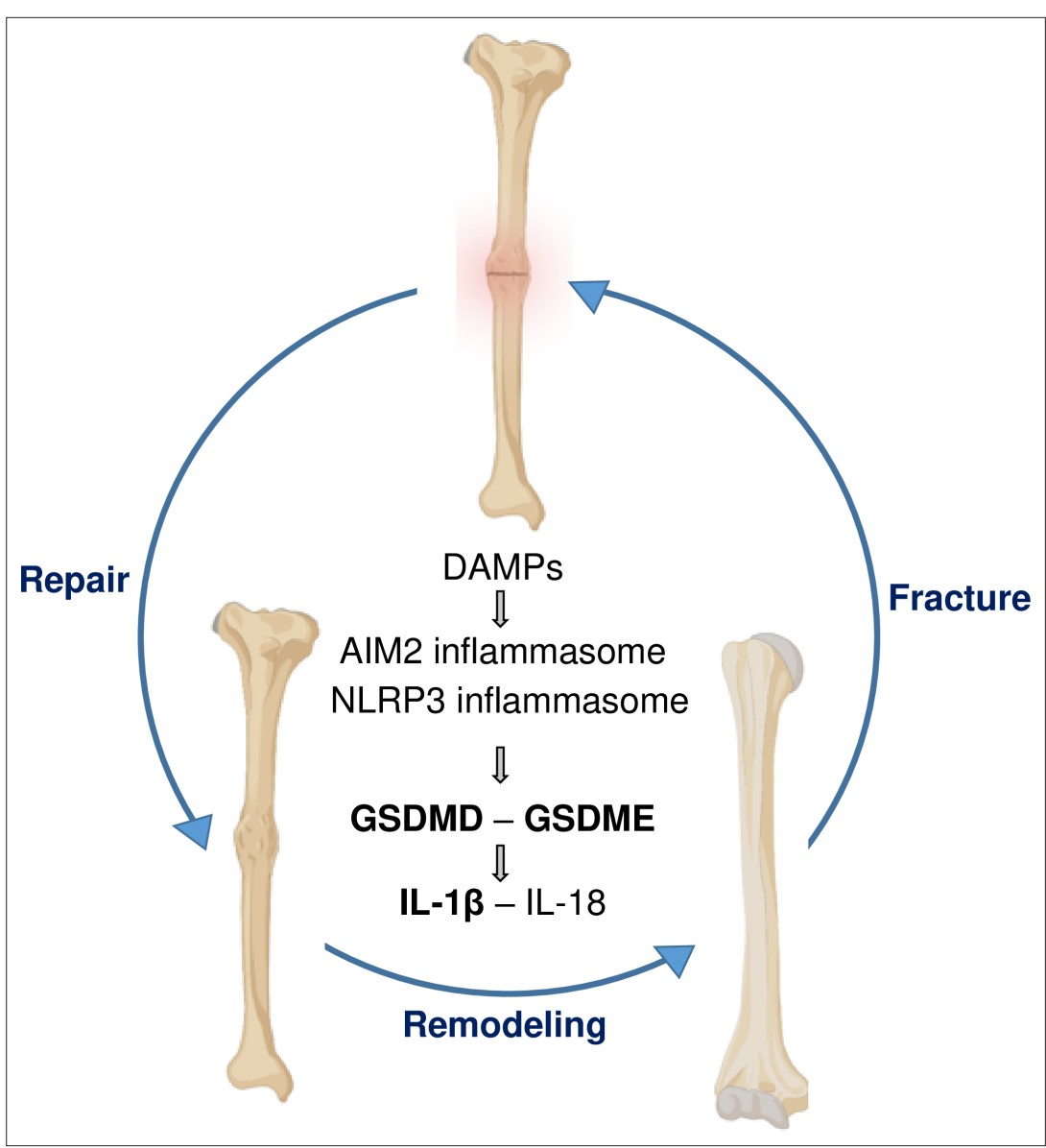

**Figure 7.** A model of gasdermin (GSDM)-interleukin-1 (IL-1) signaling in fracture healing. Bone fracture is followed by the repair and remodeling phases, which ultimately lead to the restoration of the original bone structure. Fracture causes the release of DAMPs, which activate the inflammasomes and other pathways, leading to the maturation of GSDMD and GSDME, and processing and secretion of IL-1β-IL-18. This model is based on the findings in bold texts indicating that mice lacking GSDMD, GSDME, or defective in IL-1 signaling (owing to the deletion of IL-1 receptor) exhibit delayed fracture healing. Levels of ATP (DAMP), AIM2, NLRP3, and IL-18 are elevated following fracture, but whether these factors are involved in fracture healing was not assessed in this study.

*2020*; *Blair et al., 2011*; *Hachemi et al., 2018*); (iv) the knowledge gap on differential expression of GSDMD and GSDME by the various cell types that are activated in response to fracture; and (v) the lack of translational studies using drugs such as disulfiram, which inhibits the processing or functions of GSDMD and GSDME (*Wang et al., 2021*), to validate genetic mouse findings. Despite these shortcomings, this study has revealed the crucial role that GSDMD and GSDME play in fracture healing (*Figure 7*). It also suggests that drugs that inhibit the functions of these GSDMs may have adverse effects on this healing process.

## Acknowledgements

This work was supported by NIH/NIAMS AR076758 and AI161022 grants to GM. JS was supported by NIH grants AR075860, AR077616, and AR077226; YA-A by NIH grants AR049192, AR074992, AR072623, and by grant #85160 from the Shriners Hospital for Children; MJS, by NIH grants AR074992.

## Additional information

### Competing interests

Yousef Abu-Amer: Reviewing editor, eLife. Gabriel Mbalaviele: Consultant for Aclaris Therapeutics, Inc. The other authors declare that no competing interests exist.

### Funding

| Funder | Grant reference number | Author |
|---|---|---|
| National Institute of Arthritis and Musculoskeletal and Skin Diseases | AR076758 | Gabriel Mbalaviele |
| National Institute of Allergy and Infectious Diseases | AI161022 | Gabriel Mbalaviele |
| National Institute of Arthritis and Musculoskeletal and Skin Diseases | AR075860 | Jie Shen |
| National Institute of Allergy and Infectious Diseases | AR077616 | Jie Shen |
| National Institute of Allergy and Infectious Diseases | AR077226 | Jie Shen |
| National Institute of Allergy and Infectious Diseases | AR049192 | Yousef Abu-Amer |
| National Institute of Allergy and Infectious Diseases | AR074992 | Yousef Abu-Amer |
| National Institute of Allergy and Infectious Diseases | AR072623 | Yousef Abu-Amer |
| Shriners Hospitals for Children | 85160 | Yousef Abu-Amer |
| National Institute of Arthritis and Musculoskeletal and Skin Diseases | AR074992 | Matthew J Silva |

The funders had no role in study design, data collection and interpretation, or the decision to submit the work for publication.

### Author contributions

Kai Sun, Data curation, Formal analysis, Investigation, Methodology, Validation, Visualization, Writing – review and editing; Chun Wang, Jianqiu Xiao, Tong Yang, Data curation, Formal analysis, Methodology,

Validation, Visualization; Michael D Brodt, Luorongxin Yuan, Data curation, Formal analysis, Methodology, Validation, Visualization, Writing – review and editing; Yael Alippe, Data curation, Methodology, Visualization; Huimin Hu, Dingjun Hao, Investigation, Resources; Yousef Abu-Amer, Funding acquisition, Investigation, Resources, Writing – review and editing; Matthew J Silva, Investigation, Resources, Visualization, Writing – review and editing; Jie Shen, Funding acquisition, Investigation, Methodology, Supervision, Writing – review and editing; Gabriel Mbalaviele, Conceptualization, Funding acquisition, Investigation, Project administration, Resources, Supervision, Visualization, Writing – original draft, Writing – review and editing

### Author ORCIDs
Yousef Abu-Amer http://orcid.org/0000-0002-5890-5086
Gabriel Mbalaviele http://orcid.org/0000-0003-4660-0952

### Ethics
All mice were on the C57BL6J background, and genotyping was performed by PCR. All procedures were approved by the Institutional Animal Care and Use Committee (IACUC) of Washington University School of Medicine in St. Louis. All experiments were performed in accordance with the relevant guidelines and regulations described in the IACUC- approved protocol 19-0971.

### Decision letter and Author response
Decision letter https://doi.org/10.7554/eLife.75753.sa1
Author response https://doi.org/10.7554/eLife.75753.sa2

---

# Additional files

### Supplementary files
• Transparent reporting form

### Data availability
All data generated or analyzed during this study are included in the manuscript and supporting files; source data files have been provided for all figures.

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
