## [Editor Report]

We would like to congratulate the authors on this very exciting work that will advance our understanding on the contribution of gasdermin – interleukin-1 signaling in fracture healing. We hope future work will translate this discovery into human trials.

---

## [Decision Letter]

**Decision letter after peer review:**

Thank you for submitting your article "Fracture healing is delayed in the absence of gasdermin signaling" for consideration by *eLife*. Your article has been reviewed by 2 peer reviewers, one of whom is a member of our Board of Reviewing Editors, and the evaluation has been overseen by Mone Zaidi as the Senior Editor. The following individual involved in review of your submission has agreed to reveal their identity: Carlos Isales (Reviewer #1).

Essential revisions:

1) Revise title that reflects the overall outcome of the studies presented.

2) Please take of care of the comments and the needs for clarifications (see below for specifics).

*Reviewer #1 (Recommendations for the authors):*

(1) The authors make the case that the manuscript is translationally relevant because of the potential use of Gasdermin D inhibitors in clinical settings (e.g. sepsis) however there are no experiments using these inhibitors (e.g. disulfiram).

(2) The authors seem to variably use male or female mice (Figure 1 vs 2 or 4 vs 5 for example). Was there no sex difference in any of the experiments? If not they should clearly state that and clarify in Methods section.

(3) In Figure 4 D, IL18 was the difference in WT between BM unfractured vs fractured significant?

(4) Would the authors expect the severity of the injury impact the degree of activation of GSDM? Would they expect differences in activation of GSDM depending on degree and location of injury or perhaps those resulting in bone healing vs necrosis?

(5) By eliminating inflammatory signaling (interleukin 1β) wouldn't the authors expect this to impact the anti-inflammatory pathways? For example, are glucocorticoids upregulated? Glucocorticoids themselves at lower doses can stimulate bone growth and decreasing that signal may affect bone healing and strength.

---

## [Author Response]

Reviewer #1 (Recommendations for the authors):(1) The authors make the case that the manuscript is translationally relevant because of the potential use of Gasdermin D inhibitors in clinical settings (e.g. sepsis) however there are no experiments using these inhibitors (e.g. disulfiram).

We agree with the reviewer’s comment. We have discussed this shortcoming in the revised manuscript (lines 384-385).

(2) The authors seem to variably use male or female mice (Figure 1 vs 2 or 4 vs 5 for example). Was there no sex difference in any of the experiments? If not they should clearly state that and clarify in Methods section.

We appreciate the suggestion. We have now stated in the legends of figures 1 and 2 that data from male and female were pooled because there was no sex difference. Data from the other figures were from male or female mice as specified in the original submission.

(3) In Figure 4 D, IL18 was the difference in WT between BM unfractured vs fractured significant?

No, the difference is not statistically significant.

(4) Would the authors expect the severity of the injury impact the degree of activation of GSDM? Would they expect differences in activation of GSDM depending on degree and location of injury or perhaps those resulting in bone healing vs necrosis?

This is an excellent point. We attempted to carry out studies without pin insertion to aggravate the injury, but we were unsuccessful. We will try to optimize the protocol to address this issue in future studies.

(5) By eliminating inflammatory signaling (interleukin 1β) wouldn't the authors expect this to impact the anti-inflammatory pathways? For example, are glucocorticoids upregulated? Glucocorticoids themselves at lower doses can stimulate bone growth and decreasing that signal may affect bone healing and strength.

This is another excellent point, which we have discussed in the revised manuscript (lines 380-382).